# The Assessment of Nipple Areola Complex Sensation with Semmes-Weinstein Monofilaments—Normative Values and Its Covariates

**DOI:** 10.3390/diagnostics11112145

**Published:** 2021-11-19

**Authors:** Anna Kasielska-Trojan, Agata Szulia, Tomasz Zawadzki, Bogusław Antoszewski

**Affiliations:** 1Plastic, Reconstructive and Aesthetic Surgery Clinic, Medical University of Łódź, 90-153 Łódź, Poland; tzawadzki2233@gmail.com (T.Z.); boguslaw.antoszewski@umed.lodz.pl (B.A.); 2The Military Medical Faculty, Medical University of Łódź, 90-419 Łódź, Poland; agata.szulia@stud.umed.lodz.pl

**Keywords:** breast, sensation threshold, two-point discrimination, Semmes-Weinstein monofilaments

## Abstract

Objective: To establish normative data for nipple-areola complex (NAC) sensibility examined with Semmes-Weinstein monofilament test (SWMT) and two-point discrimination (TPD) in women with varying breast sizes, including women with gigantomastia. We also aimed to identify clinical variables influencing NAC sensation. Methods: A total of 320 breasts in 160 Caucasian women (mean age 33.6 years, SD 11 years) were examined (including 50 hypertrophic breasts). NACs sensation was examined using Semmes-Weinstein monofilaments (SWM) and the Weber Two-Point Discrimination Test. Results: The nipple appeared to be the most sensitive part of NAC. In normal-sized breasts, sensation thresholds (SWM) correlated with: age, BMI, history of births, breast size and ptosis (for all locations), breastfeeding history (for nipple and upper areola) and areola diameter (for all locations apart from the nipple). Regression analysis showed that age, cup size and suprasternal notch-to-nipple distance are risk factors for diminished NAC sensation. Sensation thresholds in all NAC locations of hypertrophic breasts were significantly higher compared to normal-sized breasts, while TPD tests did not differ between the groups. Conclusions: We provided normative values of NAC sensation (tactile threshold and TPD) for different NAC areas. Our investigation indicated that SWM are useful diagnostic tools when the following factors are considered while examining NAC sensation: location (nipple vs. areola), age, breast size, suprasternal notch-to-nipple distance, history of births and breastfeeding. Hypertrophic breasts presented significantly higher sensation thresholds for all NAC locations. The report may serve as a reference data for further investigations regarding NAC sensation after different breast surgeries.

## 1. Introduction

Preservation of the sensitivity of the nipple-areola complex (NAC) remains one of the essential goals in breast surgery. Still, providing standard values of nipple and areola sensibility, which would allow for assessment and comparison of sensitivity level pre- and post-operatively, continues to pose a challenge to both researchers and clinicians. The methodology of such studies includes using Semmes-Weinstein nylon monofilaments (most commonly used) and Pressure-Specified Sensory Devices. Many researchers have reported analyses of NAC sensation after certain surgical procedures, i.e., different techniques of reduction mammaplasties (inferior or medial pedicle techniques or free nipple grafts), gender-affirming mastectomies and NAC-preserving mastectomies [1,2,3,4,5]. However, these studies were often based on pre- and post-procedure analyses or included rather small control groups and did not involve important variables which may influence NAC sensation. Identifying these variables seems to be crucial for further studies examining breast sensation after different surgical procedures involving different techniques (e.g., different pedicles in breast reduction mammoplasties) to exclude bias related to anthropometric and clinical variables. As sensation in the nipple significantly affects quality of life in women, assessment of NAC sensitivity should become a standard part of the routine evaluation of surgical outcomes. To make this assessment as reliable as possible, it seems clinically beneficial to provide ample reference data and to identify factors that influence measurements’ results.

In this study, we aimed to establish normative data for breast sensibility of the nipple-areola complex (the nipple and four quadrants of the areola) using Semmes-Weinstein monofilament tests (SWMT) and two-point discrimination (TPD) in women with varying breast sizes. We also aimed to verify the most common clinical variables as factors influencing NAC sensation (age, body mass index (BMI), breast size, NAC diameter, distance from the nipple to the suprasternal notch, history of births and breastfeeding, menstrual cycle phase, contraception—oral or intravaginal uterine system (IUS)). We hypothesize that women with larger and more ptotic breasts will present decreased NAC sensation and may lack two-point discrimination, and that the decrease will depend on breast size, the distance from sternal notch to the nipple and NAC diameter. Moreover, we expect NAC sensitivity level to depend on the phase of the menstrual cycle. In the second study, we aimed to analyze NAC sensation (SWMT and TPD) in women with gigantomastia (defined as breast hypertrophy with associated pathologic conditions resulting from excess breast weight, volume or breast malposition [6]) and determine its covariates. 

## 2. Materials and Methods

The protocol for this study was approved by the local ethics committee (of the Medical University of Lodz, RNN/366/19/KE) and all study subjects gave informed consent for sensory testing and chest measurements. The study was conducted between January 2021 and September 2021. A total number of 320 breasts in 160 Caucasian women were examined. Women with a wide spectrum of breast sizes meeting the inclusion criteria were chosen on a random basis among the patients of a gynecological outpatient clinic (study I). Additionally, we involved a group of women with diagnosed gigantomastia (study II). All women filled in a clinical questionnaire (including questions concerning: age, breast size, first day of their last menstruation to estimate the day of menstrual cycle, number of births and breastfeeding, hormonal contraception), underwent sensory testing and had breast measurements performed. Specific inclusion criteria for both studies involved: 1. women between 18 and 60 years old, 2. able to provide written consent, 3. able to provide information asked in the questionnaire, 4. ability to cooperate during the study. The exclusion criteria were: 1. pregnancy or breastfeeding and less than 6 months after the end of breastfeeding, 2. diagnosed with oncologic (previous chest radiation included), neurologic or other disorders affecting sensation in the NAC (metabolic diseases, e.g., diabetes, thyroid disorders, vascular disease, alcoholism, neurologic impairment, anemia), 3. a history of breast surgery, 4. BMI above 30 kg/m^2^. Additional inclusion criterium for study II was a diagnosis of gigantomastia defined as breast hypertrophy with associated clinical conditions caused by excess breast weight (mastalgia, neck and back pain, headaches, trophic lesions of the breast skin with ulceration and infection, limited ability to exercise) and qualification for breast reduction surgery.

### 2.1. Participants

#### 2.1.1. Study I

The main studied group consisted of 135 women (average age 33.6 years, SD 11 years) who declared their breasts as normal and not causing health problems, regardless of their size. They were recruited from patients of a gynecological outpatient clinic by the gynecologist familiar with their medical histories (consecutive patients meeting the criteria, when the researcher was available). 

#### 2.1.2. Study II

Examinations were performed in 25 women (average age 33.2 years, SD 9.3 years) who qualified for breast reduction in a Plastic, Reconstructive and Aesthetic Surgery Clinic because of symptomatic breast hypertrophy (surgery paid by National Health Fund) and met the criteria of the study. All women had breast ultrasonography or mammography examination performed to exclude breast tumors. Sensation tests were performed during preoperative visits in the out-patient clinic. 

### 2.2. Anthropometric and Sensory Measurements with SWM and Weber TPD

Sensory assessment was performed on both NACs using Semmes-Weinstein monofilaments (SWM) and the Weber Two-Point Discrimination Test (TPD). The following anthropometric measurements were taken: weight, height, suprasternal notch-to-nipple distance (sn-n) and the diameter of the NAC (or its widest measure). NAC sensitivity was assessed in five points: four quadrants of the areola and the nipple using SWMs, while two-point discrimination was evaluated in four areolar quadrants with the use of the Weber Two-Point Discrimination Test (TPD) (Exacta Touch Test, North Coast Medical, Inc., Morgan Hill, CA, USA) (Figure 1 and Figure 2). Patients were asked to sit in a temperature-controlled testing room with vision occluded during the examination [7]. Measurements of sensitivity (touch threshold) with SWMs were performed by applying force-calibrated fibers on four quadrants of the areola and the nipple of each breast and asking the patient to situate the stimulus perceived on the certain quadrant of the areola or the nipple on the right and left breast. Women feeling a stimulus located it on “the nipple” or “upper”, “lower”, “medial” or “lateral” part of “right” or “left” NACs. Examination started on the random side of the chest, from the thinnest fiber from the kit (Baseline Tactile Monofilament Cutaneous Sensory Perception Testing, Fabrication Enterprises, Inc., White Plains, NY, USA; 20 fibers; forces: 0.008 g per square millimeter (g), 0.02 g, 0.04 g, 0.07 g, 0.16 g, 0.4 g, 0.6 g, 1.0 g, 1.4 g, 2.0 g, 4 g, 6 g, 8 g, 10 g, 15 g, 26 g, 60 g, 100 g, 180 g, 300 g; filaments marked from 1.65 to 6.65—the logarithm of the force in milligrams required to bend the monofilament into a C-shape). Weber’s test was carried out by holding the points of calipers against the four quadrants of the areola at different distances from each other and determining the minimal distance at which the patient was able to determine whether one or two points were in contact with the skin of the areola. The value of 15 mm was regarded as a lack of two-point discrimination. Sensory evaluations were performed by one examiner in Study I and one examiner in Study II and were always repeated three times. In case of any discrepancies, a fourth examination was performed after at least 20 min. 

### 2.3. Statistical Analysis

The normality of distribution of each variable was tested using the Shapiro–Wilk test. In study I, we determined normative values for SWMT and TPD in different locations of NAC and checked if the analyzed variables affected their values using: Spearman rank correlation coefficients (for age, BMI, history of births, breastfeeding, breast size, sn-n distance and NAC diameter), Mann–Whitney test (for oral contraception and intrauterine system (IUS) use) and Kruskal–Wallis test (for the phase of the cycle). We also tested sensation differences between right and left breasts with a t test. In study II, we compared SWMT and TPD between women with normal breasts and women with gigantomastia with Mann–Whitney testing. According to the data provided by the manufacturer of the SWM, the cut-off value of sensation ≥0.84 indicates diminished protective sensation, which may present significant clinical value, so we designed a model (stepwise logistic regression) to determine specific variables associated with impaired sensation. 

All tests were two-tailed at a significance level of *p* < 0.05. All statistical analyses were performed using the STATISTICA package (v13, StatSoft, Cracow, Poland).

## 3. Results

### 3.1. Study I

Table 1 presents normative values of sensation thresholds and two-point discrimination in different NAC locations measured with Semmes-Weinstein monofilaments and the Weber device in normal-sized and hypertrophic breasts. Sensation thresholds differed significantly between different locations of the NAC—in normal-sized and hypertrophic breasts, the nipple appeared to be the most sensitive part of the NAC. Two-point discrimination did not differ across NAC locations in both groups, so in further analysis concerning TPD we included the mean values from all locations. Comparison of sensation thresholds and TPD did not reveal any differences between the right and left breast. 

#### 3.1.1. Factors Affecting NAC Sensation

In normal-sized breasts, sensation thresholds (SWM) significantly correlated with: age, BMI, history of births, cup size and breast ptosis (sn-n distance) (for all locations) and the history of breastfeeding (for nipple and upper areola) and the areola diameter (for all locations apart from the nipple) (Table 2). IUS (*n* = 18) significantly affected (worsened) sensation in all NAC locations; however, the positive correlation between sensation thresholds and IUS depended on a confounding factor (age). The age of women using IUS was significantly higher than that in women without the system (*p* = 0.015). Oral contraception did not affect NAC sensation (Table 3). Menstrual cycle phase or menopause did not influence SWM tests. 

The average values of TPD tests correlated with: age, BMI, history of births and breastfeeding, breast ptosis (sn-n distance) and areola diameter (Table 4). Additionally, TPD appeared to differ in the menstrual cycle; the threshold was the highest in postmenopausal women (Table 4). Contraception (oral and IUS) did not affect TPD tests. 

#### 3.1.2. Multifactorial Model (Stepwise Logistic Regression Analysis) Identifying Risk Factors of Impaired Sensation (SWM Test ≥ 3.84)

Diminished sensation was considered as the average sensation (of all NAC locations) equal to or over 3.84. The model included all clinical and anthropometric variables except for phase of the cycle, oral contraception (no association with the mean NAC sensation) and IUS (variable was related to the age). Regression analysis showed that the following variables are risk factors for impaired NAC sensation: age (increase of 1 year raises the risk 1.05 times), cup size (increase of one size raises the risk 1.76 times) and sn-n distance (with every centimeter the risk increases 1.2 times). The model correctly classified 85.6% of breasts with diminished sensation, 30.95% with sensitivity and 95.16% with specificity (Table 5).

### 3.2. Study II

Sensation thresholds of hypertrophic breasts significantly correlated with age and BMI (for upper and lower NAC locations). Sensation of medial NAC correlated with age while of lateral NAC with BMI. Contraception and cycle phase did not influence sensation thresholds. The mean values of TPD tests depended on the history of births and breastfeeding. For hypertrophic breasts, sensation thresholds in all NAC locations were significantly higher compared to those for normal-sized breasts, while TPD did not differ between the groups (Table 1).

## 4. Discussion

Preserving sensitivity of the NAC after breast surgery is the key to achieving good postoperative outcome from the aspect of patients’ body image and quality of life [8,9]. Its importance has resulted in the development of nipple-sparing operative techniques [10,11,12,13,14] as well as the modifications of breast reconstruction techniques aimed at restoring the sensation of the breast [15].

There are some studies which have examined the relation between breast size and the sensibility of the NAC [16] and changes in NAC sensitivity before and following surgical procedures [4,17,18,19]. Such studies should be referred to some normative values, and different variables which may affect the results should be considered. Tairych et al. (1998) aimed to provide a pool of data on normative NAC sensation. They studied the association between the sensitivity of the breast and its size and ptosis in 300 breasts of 150 healthy women and showed that larger breasts are significantly less sensitive than eutrophic ones and that an increase in breast ptosis correlates with a significant decrease in nipple sensibility. The authors used Regnault classification for ptosis, while in this study we used metric data—suprasternal notch-to-nipple distance [7]. We showed that the sensation thresholds of NAC positively correlated with this distance. Moreover, regression analysis showed that sn-n distance is a risk factor for diminished sensation (SWM test over 3.84) and that an increase of 1 cm raises the risk 1.2 times. Some authors also considered clinical variables, which may influence NAC sensation: age, previous pregnancies, smoking and hormonal contraception, and found significance only for previous pregnancies [7]. In our study, we found that a number of clinical and anthropometric variables influenced NAC sensation and two-point discrimination. We found that higher age and BMI, history of births and bigger cup size affected NAC sensation. For specific locations also breastfeeding (for the nipple) and areola diameter (for areolar locations) were determined as covariates of sensation thresholds. Regression analysis indicated age, breast cup size and sn-n distance as risk factors for diminished sensation.

Moreover, Tairych et al. (1998) found that the nipple was the least sensitive part of the breast [7], contrary to Mofid et al. (2002) who reported higher sensitivity for the nipple than for the areola. However, the authors included one variable, which may influence NAC sensation—breast size. They found that the nipple has approximately twice the sensitivity of the areola for both slow-adapting and fast-adapting sensory receptors. They also reported the inverse relationship between breast size and sensitivity and two-point discrimination—women with 36DD cup size or greater were found to have a greater than 10-fold decrease in sensitivity within the nipple-areola complex compared to women with normal-sized breasts (34A to 36C cup size) [12]. Our study showed that the nipple presents the lowest sensation threshold of all NAC locations and sensation of the areola depends on its diameter, which may be related to stretching of areolar nerves. Additionally, breastfeeding was a risk factor of decreased sensation in the nipple but not in most of the locations on the areola. Similar to the abovementioned study, we found that breast size was inversely correlated with NAC sensation and TPD and that hypertrophic breasts had significantly higher sensation thresholds than normal-sized breasts.

The widely reported correlation between breast size and sensation most likely results from the increased gravitational forces pulling on hypertrophic breasts which cause a stretch-type traction injury to the 2nd−6th intercostal nerves supplying the NAC area. This observation is supported by the results of follow-ups on breast reduction patients, which have shown that the reduction does not impair breast sensation, which instead remains at the same level or even increases. Additionally, the literature demonstrates that the density of sensory nerve endings mediating tactile sensibility is lower in tissues with a large infiltration of fat if compared with normal tissue, and that the removal of large amounts of fatty tissue restores sensibility [15,20]. 

Changes in NAC sensitivity during the menstrual cycle with and without the use of contraception seem to be clinically important as they may help in choosing the most optimal cycle phase for breast surgeries. The literature in this area is not extensive; however, Robinson et al. (1977) examined changes in breast sensitivity of six healthy nulliparous women over eight normal cycles and eight cycles controlled by contraceptive pills. By measuring two-point discrimination thresholds with the use of Weber discriminator and pain thresholds with SWM, they observed significant rhythmic alterations and peaks of sensitivity associated with the period directly before menstruation or with menstruation itself in female volunteers during normal menstrual cycles. During cycles regulated by oral contraceptives, the mid-cycle peak was absent in regard to touching sensitivity, and there were no changes in pain threshold sensitivity [21]. Although the etiology behind cyclical peaks in breast sensation is yet to be explained, these variations are believed to be hormonally determined. A physiological decrease in estradiol and progesterone levels [22] is associated with an increase in breast sensitivity commonly reported by women prior to or during menses [23,24]. We find these changes merit further research as they may serve as a promising factor in determining optimal timing for breast-related procedures in women. Our study showed no correlation between sensation thresholds and phase of the menstrual cycle or menopausal status. However, we did not include women in the first 4 days on their menstrual cycle as during menstruation we do not perform breast surgeries, so the clinical value of such data would be limited. IUS was found to correlate with higher sensation thresholds but it is usually used in older women with a history of births (and often breastfeeding), and the examined women using IUS were significantly older, which was the main risk factor for diminished sensation. IUS is reported to act locally but some small dosages of progestogen getting into bloodstream could act on the breast gland. This was not supported in our study but further research concerning this issue could be useful. On the other hand, endogenous hormone fluctuations during the menstrual cycle appeared not to affect sensation thresholds and TPD.

Regarding methodology, in our investigation we used Semmes-Weinstein monofilaments and a Weber two-point aesthesiometer. These are widely available and frequently utilized tools which allow for assessment of cutaneous pressure thresholds [25]. There are some drawbacks of these tools, e.g., dependence on the skill of the user and the quality and maintenance of the product [26,27]. There are data showing that normal breast sensibility using Semmes-Weinstein nylon monofilaments yields results varying by a magnitude that exceeds 10-fold and that computed devices (Pressure-Specified Sensory Devices) provide results with significantly higher inter-observer reliability [12,28]. However, there are also studies suggesting that monofilaments, if used precisely and adequately, retain the advantages of being an evidence-based, reproducible, affordable, accessible tool for assessing touching sensibility [25,27,29,30,31]. 

The study has some limitations. Firstly, we included only white women, thus the results may be ethnically specific and may not allow for generalization. Measurements were performed by two researchers; however, they were similarly experienced, and the measurements were performed three times and additionally verified, if needed. Additionally, the use of Semmes-Weinstein monofilaments and two-point discrimination aesthesiometers could have been verified with the use of Pressure-Specified Sensory Devices. The main strengths include detailed clinical characteristics of the studied sample which allowed the verification of these variables’ influence on NAC sensation.

## 5. Conclusions

Our study provided a substantial collection of data on NAC sensation in eutrophic and hypertrophic breasts and presented normative values of sensation and two-point discrimination for different areas of NAC. Our investigation indicated that SWM are useful diagnostic tools when the following factors are considered while examining NAC sensation in clinical studies: age, breast size, suprasternal notch-to-nipple distance, history of births and breastfeeding. Age, cup size and sn-n distance are risk factors for diminished NAC sensation. Hypertrophic breasts presented significantly higher sensation thresholds for all NAC locations. This report may provide reference data for further investigations regarding NAC sensation after different breast surgeries.

## Figures and Tables

**Figure 1 diagnostics-11-02145-f001:**
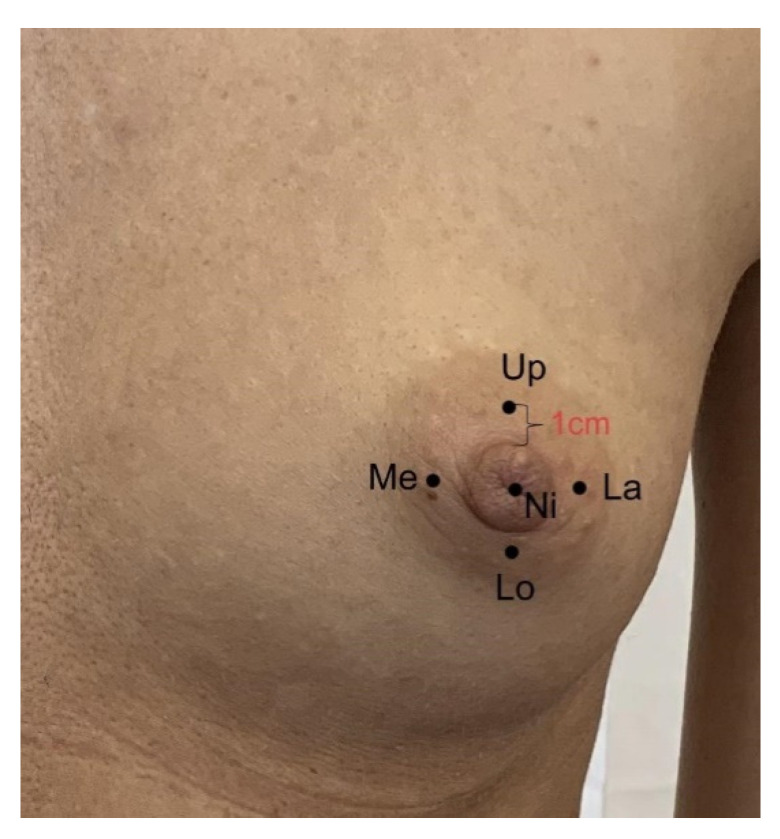
Five NAC areas measured for sensation thresholds with SWMs. The measured points 1 cm from the nipple margin were labeled: Ni = nipple, Up = upper, Lo = lower, Me = medial, La = lateral.

**Figure 2 diagnostics-11-02145-f002:**
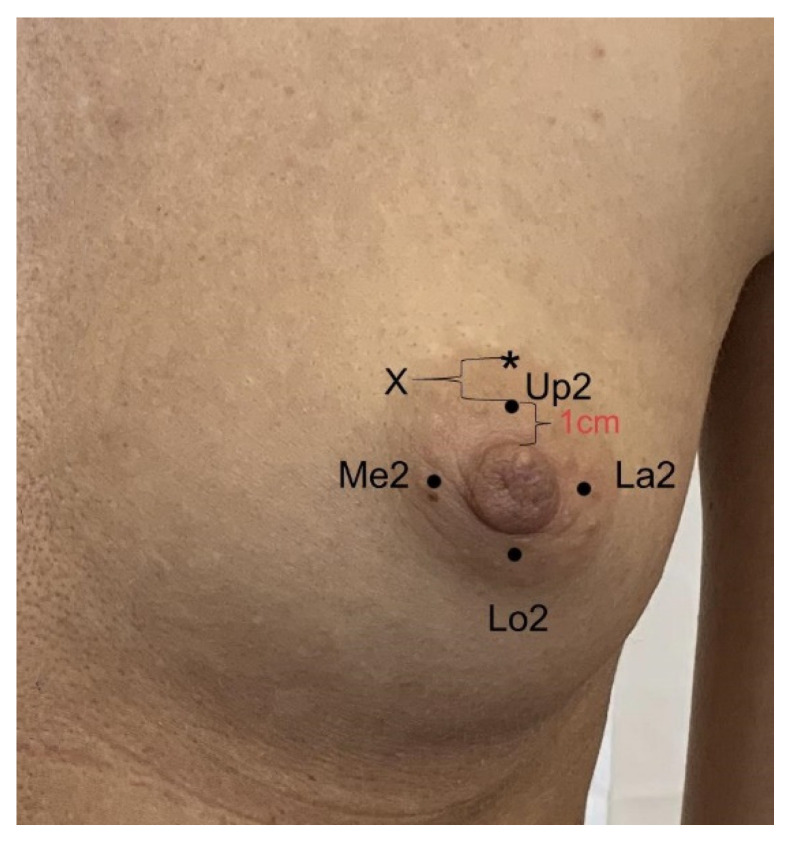
Four NAC areas measured for two-point discrimination with Weber esthesiometer. The most central points of esthesiometer placed 1 cm from the nipple margin were labeled: Up2 = upper, Lo2 = lower, Me2 = medial, La2 = lateral. *—distal point of esthesiometer arm when two-point touch was detected, X—the result of examination.

**Table 1 diagnostics-11-02145-t001:** Normative values of sensation thresholds and two-point discrimination in different NAC locations measured with Semmes-Weinstein monofilaments and Weber device in normal-sized and hypertrophic breasts and comparison of these variables between groups.

	Normal-Sized Breasts *n* = 270					Hypertrophic Breasts *n* = 50					Normal vs. Hypertrophic
	Median	Quartiles	Mean	SD	Chi/*p*	Median	Quartiles	Mean	SD	Chi/*p*	*p* **
SWMT					79.96 *p* < 0.0001	SWMT				47.81 *p* < 0.0001	
nipple	3.22	2.83; 3.61	3.03	0.59		3.22	2.83; 3.61	3.24	0.42		0.013
upper	3.61	2.83; 3.84	3.23	0.77		3.61	3.22; 4.08	3.55	0.46		0.027
medial	3.22	2.83; 3.84	3.26	0.74		3.61	3.22; 4.08	3.55	0.45		0.038
lower	3.22	2.44; 3.84	3.21	0.82		3.61	3.22; 4.08	3.67	0.47		0.001
lateral	3.22	2.44; 3.84	3.19	0.82		3.61	3.22; 4.08	3.61	0.47		0.003
Mean *	3.22	2.75; 3.69	3.18	0.60		3.56	3.3; 3.77	3.52	0.36		0.0003
	2PDT				4.95 *p* = 0.176	2PDT				1.20 *p* = 0.753	
upper	15	13; 15	13.66	2.39		15	12; 15	13.32	2.54		0.357
medial	15	13; 15	13.67	2.42		15	12; 15	13.32	2.83		0.931
lower	15	14; 15	13.80	2.20		15	12; 15	13.26	2.82		0.39
lateral	15	13; 15	13.71	2.30		15	12; 15	13.26	2.98		0.829
Mean *	14.75	13.75; 15	13.71	2.14		14.75	12.75; 15	13.29	2.62		0.441

* mean value for all NAC locations, ** Mann–Whitney test.

**Table 2 diagnostics-11-02145-t002:** Influence of clinical and anthropometric variables on sensation thresholds in different NAC locations measured with Semmes-Weinstein monofilaments in normal-sized breasts.

Normal-Sized Breasts *n* = 270	Nipple		Upper		Medial		Lower		Lateral		Mean *	
	R **	*p*	R	*p*	R	*p*	R	*p*	R	*p*	R	*p*
age	0.30	<0.0001	0.25	<0.0001	0.22	0.0003	0.16	0.008	0.24	<0.0001	0.28	<0.0001
BMI	0.22	0.0002	0.32	<0.0001	0.35	<0.0001	0.30	<0.0001	0.39	<0.0001	0.39	<0.0001
births	0.25	<0.0001	0.18	0.003	0.14	0.0182	0.06	0.306	0.13	0.03	0.18	0.003
breastfeeding	0.24	<0.0001	0.16	0.009	0.10	0.091	0.03	0.655	0.11	0.083	0.14	0.02
cup size	0.23	0.0001	0.34	<0.0001	0.42	<0.0001	0.37	<0.0001	0.41	<0.0001	0.45	<0.0001
sn-*n*	0.25	<0.0001	0.43	<0.0001	0.47	<0.0001	0.42	<0.0001	0.44	<0.0001	0.49	<0.0001
areola diameter	0.09	0.153	0.40	<0.0001	0.41	<0.0001	0.37	<0.0001	0.34	<0.0001	0.39	<0.0001

* mean value for all NAC locations. ** Spearman R correlation coefficient.

**Table 3 diagnostics-11-02145-t003:** Contraception influence on sensation thresholds in different NAC locations measured with Semmes-Weinstein monofilaments in normal-sized breasts.

Normal-Sized Breasts *n* = 270	Oral Contraception = 36				No Oral Contraception *n* = 234				*p* **
	Median	Quartiles	Mean	SD	Median	Quartiles	Mean	SD	
nipple	2.83	2.44; 3.22	2.92	0.64	3.22	2.83; 3.61	3.05	0.59	0.176
upper	3.22	2.635; 3.84	3.13	0.80	3.61	2.83; 3.84	3.25	0.77	0.401
medial	3.22	2.635; 3.725	3.17	0.77	3.22	2.83; 3.84	3.27	0.74	0.482
lower	3.22	2.44; 3.61	3.01	0.83	3.61	2.44; 3.84	3.24	0.82	0.109
lateral	2.44	2.36; 3.61	2.92	0.88	3.61	2.44; 3.84	3.23	0.80	0.047
mean *	3.04	2.594; 3.478	3.03	0.56	3.29	2.814; 3.702	3.21	0.61	0.069

* mean value for all NAC locations, ** Mann–Whitney test.

**Table 4 diagnostics-11-02145-t004:** Influence of clinical and anthropometric variables on mean (for all NAC locations) two-point discrimination thresholds measured with Weber device in normal-sized breasts.

Normal-Sized Breasts *n* = 270	R *	t	*p*
age	0.29	5.02	<0.0001
BMI	0.24	3.96	<0.0001
births	0.30	5.21	<0.0001
breastfeeding	0.32	5.55	<0.0001
cup size	0.05	0.77	0.443
sn-n	0.23	3.87	0.0001
areola diameter	0.28	4.82	<0.0001
Cycle phase	Mean (SD)	H	*p* **
follicular	13.59 (2.26)	10.66	0.014
ovulation	13.83 (2.35)		
luteal	13.64 (2.12)		
menopause	14.82 (0.60)		

* Spearman R correlation coefficient, ** Kruskal–Wallis test.

**Table 5 diagnostics-11-02145-t005:** Stepwise logistic regression (multifactorial model) for the risk of diminished NAC sensation.

*n* = 270	OR	95% CI	Wald Chi-Square	*p*
age	1.05	1.02; 1.09	8.03	0.005
size	1.78	1.21; 2.62	8.56	0.003
sn-n	1.2	1.06; 1.36	8.81	0.003

Model’s sensitivity—30.95%, specificity—95.16%.

## Data Availability

Data available on request from the corresponding author.

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
