# Peer review of "The Assessment of Nipple Areola Complex Sensation with Semmes-Weinstein Monofilaments—Normative Values and Its Covariates"

_diagnostics, 2021, doi:10.3390/diagnostics11112145_

Round 1
Reviewer 1 Report
The manuscript is of high quality and very interesting. However, the writing is challenging at times and several sections are difficult to read. I detected problems in the following lines of the manuscript: 83, 88, 98-104, 106-110, 119-125, 167-171, 191-193, 196, 222-224 and 237. Similarly, the data tables are hard to follow and should be simplified for the benefit of the reader. Lastly, Table 1 is nearly identical to Table 6 and should be consolidated.
Author Response
Dear Editor and Reviewers,
Thank you for your interest in our manuscript entitled " The assessment of nipple areola complex sensation with Semmes-Weinstein monofilaments – normative values and its covariates”. We would like to thank anonymous Reviewers for their valuable comments and reviews, which helped to improve the manuscript. In this revision we addressed all your comments (in the text changes are marked in red colour or described in comments in Track Changes). We hope that this revision meets with your approval.
Sincerely,
The Authors
Responses to Reviewers’ comments:
Reviewer 1:
- The manuscript is of high quality and very interesting. However, the writing is challenging at times and several sections are difficult to read. I detected problems in the following lines of the manuscript: 83, 88, 98-104, 106-110, 119-125, 167-171, 191-193, 196, 222-224 and 237.
Indicated “problematic” sections were revised – thank you for pointing on them.
- Similarly, the data tables are hard to follow and should be simplified for the benefit of the reader.
Due to the “numeric” nature of the data, it was difficult to simplify the tables, but agreeing with the remark of the Reviewer, we decided to remove some data (e.g. tests’ values, medians in table 4 and keep only means, reduce decimals) to make the tables more readable for the Readers, also we found some typos in table 2, which were corrected.
- Lastly, Table 1 is nearly identical to Table 6 and should be consolidated.
Table 1 was consolidated with table 6.
Reviewer 2:
- Abstract: consider reworking it in order to make it more focused on the aims and endpoints of the study. Furthermore, it is too long, try to reduce it by 25%.
The Abstract was revised and reduced to 250 words.
- Page 2, line 67: The authors should specify how they define “breast hypertrophy”, since it is considered a main feature for inclusion in the second study.
It was specified in the pointed section that: “Additional inclusion criterium for study II was a diagnosis of gigantomastia defined as breast hypertrophy with associated clinical conditions caused by excess breast weight (mastalgia, neck and back pain, headaches, trophic lesions of the breast skin with ulceration and infection, limited ability to exercise) and qualification for breast reduction surgery.”
- The authors should at least mention other studies conducted with Pressure-Specified Sensory Devices, that has been deemed superior in esthesiometric assessment (PMID: 32128706; PMID: 12045551).
The following reference was added:
Longo B, Timmermans FW, Farcomeni A, Frattaroli JM, D'orsi G, Atzeni M, Sorotos M, Laporta R, Santanelli di Pompeo F. Septum-Based Mammaplasties: Surgical Techniques and Evaluation of Nipple-Areola Sensibility. Aesthetic Plast Surg. 2020 Jun;44(3):689-697. doi: 10.1007/s00266-020-01657-7. Epub 2020 Mar 3. PMID: 32128706.
The second suggested reference (Mofid MM, Dellon AL, Elias JJ, Nahabedian MY. Quantitation of breast sensibility following reduction mammaplasty: a comparison of inferior and medial pedicle techniques. Plast Reconstr Surg. 2002 Jun;109(7):2283-8. doi: 10.1097/00006534-200206000-00018. PMID: 12045551) was cited in the initial version, and referenced in the section regarding PSSD superiority.
Reviewer 2 Report
Congratulation to the authors for their study on the assessment of nipple areola complex sensation. The manuscript is detailed and meticulous. Some minor critiques to the content of the manuscript should be addressed:
- Abstract: consider reworking it in order to make it more focused on the aims and endpoints of the study. Furthermore, it is too long, try to reduce it by 25%.
- Page 2, line 67: The authors should specify how they define “breast hypertrophy”, since it is considered a main feature for inclusion in the second study.
- The authors should at least mention other studies conducted with Pressure-Specified Sensory Devices, that has been deemed superior in esthesiometric assessment (PMID: 32128706; PMID: 12045551).
Author Response

(The authors gave the same response as above.)
